# A Theoretical Study on the Degradation Mechanism, Kinetics, and Ecotoxicity of Metronidazole (MNZ) in •OH- and SO_4_^•−^-Assisted Advanced Oxidation Processes

**DOI:** 10.3390/toxics11090796

**Published:** 2023-09-20

**Authors:** Jingyu Sun, Ruijun Chu, Zia Ul Haq Khan

**Affiliations:** 1Hubei Key Laboratory of Pollutant Analysis & Reuse Technology, College of Chemistry and Chemical Engineering, Hubei Normal University, Cihu Road 11, Huangshi 435002, China; 15845304247@163.com; 2Department of Chemistry, COMSATS University Islamabad, Park Road, Islamabad 45550, Pakistan; ziaulhaqkhan11@gmail.com

**Keywords:** metronidazole (MNZ), quantum chemical calculations, degradation mechanism, rate constants, ecotoxicity assessment

## Abstract

Metronidazole (MNZ), a typical example of nitroimidazole antibiotics, is widely used in the treatment of infectious diseases caused by anaerobic bacteria. The degradation mechanism and kinetics of MNZ in the presence of HO• and SO_4_^•−^ were studied using density functional theory (DFT). It was confirmed that both HO• and SO_4_^•−^ easily added to the carbon atom bonded to the NO_2_ group in the MNZ molecule as the most feasible reaction channel. This study shows that subsequent reactions of the most important product (M-P) include the O_2_ addition, hydrogen abstraction and bond breakage mechanisms. The rate constants of HO• and SO_4_^•−^-initiated MNZ in the aqueous phase were calculated in the temperature range of 278–318 K. The total rate constants of MNZ with HO• and SO_4_^•−^ were determined to be 8.52 × 10^9^ and 1.69 × 10^9^ M^−1^s^−1^ at 298 K, which were consistent with experimental values of (3.54 ± 0.42) × 10^9^ and (2.74 ± 0.13) × 10^9^ M^−1^s^−1^, respectively. The toxicity of MNZ and its degradation products to aquatic organisms has been predicted. The results proposed that the toxicity of the initial degradation product (M-P) was higher than that of MNZ. However, further degradation products of MNZ induced by HO• were not harmful to three aquatic organisms (fish, daphnia, and green algae). This study provides a comprehensive theoretical basis for understanding the degradation behavior of MNZ.

## 1. Introduction

The general use of antibiotics in human and veterinary medicine indirectly introduces their traces into the water and soil environments [1,2]. Metronidazole (MNZ) is a common type of anti-anaerobic and anti-protozoal antibiotic [3]. It is extensively used as an additive in poultry and fish feeds [4]. In addition, MNZ is an effective radiosensitizer which can be used in radiotherapy to overcome the radio-resistance of hypoxic tumor cells [5,6,7]. Due to its widespread use, MNZ has been detected in drinking water, ground water, surface water, waste water, and soil sediment at a range from one to several ng L^−1^ [8]. In order to investigate the environmental effect of MNZ, removal of MNZ from the water environment is necessary.

Different degradation techniques like adsorption [9], biodegradation [10], and advanced oxidation processes (AOPs) [4,11,12], including UV, UV/H_2_O_2_, Fenton reaction, granular activated carbon (GAC)/persulfate (PS), and gamma radiolysis, have been applied to remove metronidazole from wastewater. The literature has shown that advanced oxidation processes show good performance in the removal of antibiotics [13,14,15]. The AOPs based on gamma-ray, X-ray, and electron beams from the electron accelerators are also very important techniques which are broadly applied in the decomposition of various pollutants in water and waste water [16]. It has been reported that the UV/peroxide (UV/H_2_O_2_) and UV/persulfate (UV/PS) have better degradation efficiencies for some pharmaceuticals [17,18]. During the advance oxidation process (UV/H_2_O_2_) and (UV/PS), highly reactive species, i.e., HO• and SO_4_^•−^ radicals, are generated, initiating the degradation of antibiotics [19,20]. The HO• radicals are very reactive and react rapidly with many pharmaceuticals, with a rate constant of 10^8^–10^10^ M^−1^ s^−1^ [21]. The redox potential of SO_4_^•−^ is 2.5–3.1 V, which is higher than that of HO• (1.9–2.7 V) [22]. Thus, AOPs based on SO_4_^•−^ are gaining more interest [23,24,25,26]. Wojnárovits tabulated and discussed the rate constants of SO_4_^•−^ radicals with about 230 organic molecules. They found that the highest rate constants for SO_4_^•−^ radicals with aromatic molecules were in the ~10^9^ M^−1^s^−1^ range [27]. 

Shemer et al., in 2006, investigated the degradation rates and removal efficiencies of MNZ through advance oxidation process (UV, UV/H_2_O_2_, H_2_O_2_/Fe^2+^, and UV/H_2_O_2_/Fe^2+^) in de-ionized water [28]. Ammar et al. studied sono-Fenton and solar photo-Fenton processes and found that they were effective for the degradation of MNZ [29,30]. Cheng et al. also investigated the degradation of MNZ through advanced oxidation processes [31]. Kinetic models for fifteen pharmaceuticals and personal care products (PPCPs), including metronidazole, were built to predict the degradation of PPCPs in the advanced oxidation processes (AOPs) of HO• and SO_4_^•−^. The secondary reaction rate constants of metronidazole with HO• and SO_4_^•−^ radicals were (3.54 ± 0.42) × 10^9^ and (2.74 ± 0.13) × 10^9^ M^−1^ s^−1^, respectively [32]. Whillans et al. measured the rate constant for HO• with MNZ by means of the pulse radiolysis technique, and its value was (5.5 ± 0.2) × 10^9^ M^−1^ s^−1^ [33]. These studies were based on advanced oxidation processes which mainly focused on accelerating the degradation of MNZ by optimizing the reaction conditions and analyzing the kinetics, but without providing the specific MNZ degradation mechanism or products.

Recently, only one study has obtained the main cleavage products of MNZ based on DFT using a statistical molecular fragmentation (SMF) model [34]. However, the detailed reaction potential energy surfaces for •OH-initiated MNZ degradation were not provided. Yao et al. used DFT calculations to give the detailed degradation pathways and possible degradation products for the reactions of dimetridazole (DMZ) and ornidazole (ONZ) with HO• [35], which is helpful in exploring the degradation of MNZ initiated by HO•. Recently, quantum chemical calculations have become an effective method by which to investigate the mechanism and kinetics of HO• and SO_4_^•−^ with some organic contaminants [36,37,38,39,40]. For example, the degradation mechanism and products of acetaminophen and *N*,*N*-diethyl-m-toluamide (DEET), initiated by HO• and SO_4_^•−^ in the aqueous phase, have been studied using the quantum chemical method by Xu et al. [38,39]. Theoretical studies give us more information regarding the structures, thermodynamic data, and microscopic reaction mechanism.

To our knowledge, there have been no theoretical studies on MNZ with HO• and SO_4_^•−^. Thus, in the current paper, the quantum chemical method was used to investigate the degradation mechanism of HO• and SO_4_^•−^ with MNZ in aqueous solution reactions. On the basis of the reaction mechanism, the rate constants of all initial reaction pathways were calculated in the temperature ranges of 278–318 K. In addition, the ecotoxicity of MNZ and its degradation products to aquatic organisms was predicted in order to assess their exposure risk.

## 2. Computational Methods

### 2.1. Electronic Structure and Kinetic Calculations

The geometric optimization of all structures was performed at the M06-2X/6-31+G(d, p) level using Gaussian 16 software [41]. Previous studies have confirmed the reliability of the density functional M06-2X method for optimizing the geometries of transition states, intermediates, and products involved in the reactions of nitroimidazole drugs with HO• [34,35]. To obtain more accurate energies, single-point energies were calculated using the more expensive 6-311++G(3df, 2p) basis set. In addition, intrinsic reaction coordinate (IRC) calculations [42] were performed to verify whether each transition state was connected to the corresponding reactants and products. The solvent effect was considered using a universal solvation model of SMD [43] that includes the bulk electrostatic contribution and the cavity-dispersion-solvent-structure term. The SMD has been successfully applied in some aqueous reactions involving HO• and SO_4_^•−^ [38,39,44]. Based on the thermodynamic properties of these reactions, the bimolecular reaction rate constants in the aqueous phase were performed using the conventional transition state theory (TST) [45], with the Wigner tunneling correction [46], using the KiSThelP program [47]. The calculated formula is given below.
(1)k=κ(T)σkBThRTP0Δne−ΔG≠kBT
(2)$$\Delta G^{\neq }=G_{TS}-G_{R}$$

In the formula, *k* is the calculated rate constant. *k*_B_, *h* and *R* are the Boltzmann constant, Planck constant, and gas constant, respectively. The Gibbs free energy of activation (Δ*G^≠^*) was equal to the Gibbs free energy of the transition state minus the total Gibbs free energy of the reactants. *σ* represents the reaction path degeneracy, which accounts for the number of equivalent reaction paths. *P*^0^ and *T* are pressure and temperature, respectively. Δ*n* was calculated to be (*n* − 1) for the n-molecular reaction. For instance, Δ*n* was 1 for the bimolecular reaction studied herein. The tunneling correction factor *κ*(*T*) was calculated using the Wigner model:(3)$$\kappa (T)=1+\frac{1}{24}{\left(\frac{h\mathrm{lm}(V^{\neq })}{k_{b}T}\right)}^{2}$$
where lm(*V*^≠^) is the transition state’s imaginary frequency. Some studies have used the KiSThelP program to calculate rate constants, with good results [48,49,50]. 

### 2.2. Ecotoxicity Assessment

In this research work, the ecotoxicity levels of MNZ and its degradation products were measured by adopting the Ecological Structure–Activity Relationship Model (ECOSAR V2.0) [51]. The procedure measured the acute and chronic toxicity to three aquatic organisms, including green algae, daphnia, and fish, as a basis for assessment. The median lethal concentration (LC50) was used to calculate the acute toxicity for fish and daphnia. Median effect concentration (EC50) was used to calculate the acute toxicity for green algae. The chronic toxicity value (ChV) was used to reflect the chronic toxicity for three aquatic organisms. The lowest toxicity value was chosen as the most conservative estimate.

## 3. Results and Discussion

Figure 1 shows the optimized structure of MNZ, HO•, and SO_4_^•−^. The geometries of the transition states involved in the initial reactions of HO• and SO_4_^•−^ are presented in Figure 2. The reactions of MNZ with HO• and SO_4_^•−^ can be classified into the following types: (a) radical adduct formation (RAF) at the C1, C2, C3 and N1 of imidazole ring (RAF-1 to RAF-4); (b) hydrogen atom transfer (HAT) on the imidazole ring (HAT-1 and HAT-6) and in the methyl group (HAT-2 and HAT-7) and the –CH_2_CH_2_OH group (HAT-3 to HAT5 and HAT-8 to HAT-10). The intramolecular hydrogen bond would appear in some transition states, as indicated by Qu et al. [52]. 

### 3.1. Initial Reaction of MNZ with HO• and SO_4_^•−^

#### 3.1.1. Radical Adduct Formation Reactions

Figure 3 shows that HO• and SO_4_^•−^ could add to the –C=C– and –C=N bonds of the imidazole ring to form intermediates (IM1 to IM3 and IM1-S to IM3-S) and products (M-P and M-P-S). Furthermore, the thermodynamic data of all channels were calculated and were listed in Appendix A. The addition of both HO• and SO_4_^•−^ to the imidazole ring was exothermic, except for when added to the N1 atom of the imidazole ring. Meanwhile, HO• and SO_4_^•−^ addition to the N1 atom of the imidazole ring resulted in the highest Gibbs energy barriers, 26.92 and 37.12 kcal mol^−1^, respectively, which possibly have been associated with the greater electronegativity of the N atom. The C2 and C3 addition reactions for HO• and SO_4_^•−^ had relatively lower barriers and higher exothermicity. It was notable that HO• and SO_4_^•−^ were added to the C1 position in the MNZ molecule with the lowest Gibbs free energy barriers of 5.89 and 6.78 kcal mol^−1^, respectively, releasing 29.01 and 24.06 kcal mol^−1^. The addition of HO• and SO_4_^•−^ caused the removal of the –NO_2_ group (RAF-1 and RAF-5) and generated by-products like the hydroxyimidazole compound (M-P) and the sulfate anion substance (M-P-S). As shown in Figure 2, the bond lengths of the oxygen atoms in the HO• and SO_4_^•−^ radicals approaching C1 atom were 2.085 and 1.911 Å in RAF-1TS and RAF-5TS, respectively. Meanwhile, the C–N bond lengths in RAF-1TS and RAF-5TS were 1.429 and 1.456 Å, which exceeded MNZ by 0.023 and 0.050 Å. The HO• radicals with dimetridazole (DMZ) and ornidazole (ONZ) caused the addition to the C1 position, followed by the products of D-P/O-P and NO_2_, with barriers of 4.94 and 6.32 kcal mol^−1^, respectively [35], which were the most important reaction channels. The same mechanisms were suggested for the reaction of HO• with MNZ, DMZ, and ONZ. However, the results obtained by Xu et al. showed that nitro-elimination reactions included two steps where a single HO• or double HO• was added, then –NO_2_ was leaved [34]. Therefore, M-P as a representative product was selected in order to discuss its subsequent reactions.

#### 3.1.2. Hydrogen Abstraction Reactions

In the structure of MNZ, there are five different hydrogen-involved imidazole ring and branching chains. Intermediates (IM4–IM8) and H_2_O were generated by five hydrogen atom abstraction reaction pathways of MNZ with HO•. They were all exothermic reactions (Δ*G* = −3.08 to −30.76 kcal mol^−1^), with Δ*G*^≠^ values of 8.33 to 43.77 kcal mol^−1^. For the reactions of SO_4_^•−^ and MNZ, hydrogen abstraction from imidazole ring (HAT-6) via a higher Gibbs free energy barrier of 42.26 kcal mol^−1^ was endothermic by 5.37 kcal mol^−1^. However, the others were both exothermic (Δ*G* = −6.45 to −22.31 kcal mol^−1^), with Gibbs free energy barriers of 12.00–31.88 kcal mol^−1^. From the calculated thermodynamic data, for either HO• or SO_4_^•−^, it was confirmed that the ordering of the difficulty of the hydrogen abstraction in the MNZ molecule could be derived as: HAT-1(HAT-6) > HAT-5(HAT-10) > HAT-3(HAT-8) > HAT-2(HAT-7) ≈ HAT-4(HAT-9). It was noted that the H attached to the C2 atom of the imidazole ring was the most inaccessible for HO• and SO_4_^•−^ abstraction. However, hydrogen abstraction from the C6 atom in the –CH_2_OH branched ring was the most important, because the relative free energies were 8.33 and 12.00 kcal mol^−1^, and they were exothermic, 21.12 and 12.67 kcal mol^−1^, for the HO• and SO_4_^•−^ reactions, respectively. The abstraction of hydrogen on C atom in the –CH_2_OH group was investigated by Xu et al. [34]. They calculated that this step needed to overcome a barrier of 5.3 kcal mol^−1^. In HAT-4TS and HAT-9TS, the breaking bond (C–H) lengths were 1.151 and 1.148 Å, and the forming O–H bond lengths were 1.554 and 1.610 Å for HO• and SO_4_^•−^ hydrogen abstraction, respectively.

All the hydrogen abstraction pathways were required to overcome many more energy barriers than the RAF-1 pathway, which is consistent with the reaction mechanism of dimetridazole (DMZ) and ornidazole (ONZ) with HO• radicals. Therefore, for the initial reaction pathways of MNZ with HO• and SO_4_^•−^ radicals, addition at the position of C1 in the MNZ molecule was the most feasible reaction channel. The reaction mechanism of MNZ degraded by HO• and SO_4_^•−^ was able to guide other new nitroimidazoles to be removed, which represents the novelty and significance of this study.

In order to investigate the products of MNZ degradation in wastewater in detail, the secondary reaction pathways of M-P obtained from reacting MNZ with HO• were investigated at the double level of M06-2X/6-31+G(3df, 2p)//M06-2X/6-31+G(d, p).

#### 3.1.3. Further Conversion Reactions of M-P

Based on the above results, we concluded that M-P is the most important degradation product for the initial reaction of MNZ with HO•. And M-P can undergo a series of subsequent transformation reactions to produce other degradation products. Therefore, it is essential to study the subsequent reactions of this product in order to assess the risk of these products in an aquatic environment.

The Fukui function based on DFT was successfully adopted in order to predict the attacking site of the reactant molecules [53,54,55]. For the Fukui function, the following three situations were as follows:$$\mathrm{Nucleophilic}\;\mathrm{attack}:\;f_{k}^{+}=q_{k}(N+1)-q_{k}(N)$$
$$\mathrm{Electrophilic}\;\mathrm{attack}:\;f_{k}^{-}=q_{k}(N)-q_{k}(N-1)$$
$$\mathrm{Radical}\;\mathrm{attack}:\;f_{k}^{0}=\frac{1}{2}\left[q_{k}(N+1)-q_{k}(N-1)\right]$$
where *f_k_*^+^, *f_k_*^−^ and *f_k_*^0^ represent nucleophilic, electrophilic, and radical attacks, respectively. *q_k_*(*N* + 1), *q_k_*(*N*) and *q_k_*(*N* − 1) are the electronic populations of the (*N* + 1), *N*, and (*N* − 1) electron states, respectively. A high concentrated Fukui index (*f*^0^) indicates the most likely site to be attacked by free radicals. As shown in Appendix A, the C1 site had the largest Fukui index (*f*^0^) value of 0.1038. Thus, the C1 site is the most favorable attack site for the subsequent reaction of M-P.

Based on the Fukui function prediction, HO• was preferentially added to the C1 site in the M-P molecule. The Gibbs free energy barrier of 2.09 kcal mol^−1^ needed to be overcome in order to obtain IM9. As shown in Figure 4, the IM9 was able to perform three types of reactions: (I) branched chain breakage; (II) intramolecular H-transfer and C–C bond breakage; and (III) reaction with O_2_. The corresponding transition states were shown in Appendix A. Through comparing the thermodynamic data of the three reactions, it was revealed that P1 and IM10, as unstable degradation products, were obtained from the branched chain breakage and intramolecular H transfer, accompanied with C–C bond breakage. The above two types of reactions were endothermic, and their Gibbs free energy barriers (Δ*G*^≠^) were 31.36 kcal mol^−1^ and 36.06 kcal mol^−1^, respectively. For the reaction of IM9 with O_2_, IM9 firstly reacted with O_2_ via H abstraction from the C–H/O–H bond to produce P2, P3, and HO_2_. The P2 and P3 formation were exothermic by 29.40 and 37.95 kcal mol^−1^. P1 and P2 are double •OH imidazole products that have been found in subsequent reactions of •OH with DMZ and ONZ. It is important that the O_2_ can associate with the C atom of an unpaired electron of IM9 from the anti- or syn-position with respect to the •OH. The peroxyl radicals IM11 and IM12 were generated from spontaneous exothermic reactions, with the corresponding Δ*G* = −25.19 kcal mol^−1^ and Δ*G* = −26.15 kcal mol^−1^ between IM9 and O_2_, respectively. Subsequently, the toxicity of the final products was investigated and analyzed.

### 3.2. Kinetic Study

The bimolecular rate constants of the elementary aqueous reactions induced by HO• and SO_4_^•−^ for MNZ were calculated using transition state theory (TST) in a temperature range from 278 to 318 K, as shown in Appendix A. Aqueous phase temperatures of 278–318 K were considered in order to avoid ice and water vapor. The rate constants in the solid phase were difficult to be calculated, as reactions involving ice are extremely complex and different from reactions in the gas and aqueous phases. The transition state theory is no longer applicable for calculating the solid phase rate constant. However, the reaction rate constants involving water vapor could be calculated by TST. Previous studies have shown that the kinetic calculations for gas-phase reactions could provide better results [40,49,54]. And gas-phase rate constants could be changed to aqueous-phase ones. However, in this paper, gas-phase rate constants were not calculated, because the reactions of MNZ with HO• and SO_4_^•−^ mainly occurred in the aqueous phase. The calculated reaction rate constants and branching ratio of MNZ with HO• and SO_4_^•−^ were summarized in Table 1. The total rate constant of the aqueous phase oxidation reaction of HO• with MNZ at 298 K was 8.52 × 10^9^ M^−1^ s^−1^, in agreement with the experimental rate constants of (3.54 ± 0.42) × 10^9^ M^−1^ s^−1^ and (5.5 ± 0.2) × 10^9^ M^−1^ s^−1^ [32,33]. In our previous study, the calculated rate constants for DMZ + HO• and ONZ + HO• reactions were 4.32 × 10^9^ M^−1^ s^−1^ and 4.42 × 10^9^ M^−1^ s^−1^, respectively, which were also close to the experimental values (3.65 ± 0.01) × 10^9^ and (2.66 ± 0.26) × 10^9^ M^−1^ s^−1^, respectively. It is indicated that kinetics of the three reactions would be influenced by the substituent groups (–CH_2_CH_2_OH, –CH_3_, –CH_2_CHOHCH_2_Cl). The calculated total rate constant initiated by SO_4_^•−^ was 1.69 × 10^9^ M^−1^ s^−1^ at 298 K, which was consistent with a previous reported value (2.74 ± 0.13) × 10^9^ M^−1^ s^−1^ [32]. It was indicated that the degradation efficiencies of HO• and SO_4_^•−^ for MNZ were essentially equal throughout the AOPs.

Figure 5 shows the rate constants of MNZ reaction, with HO• and SO_4_^•−^ as a function of temperature from 278 to 318 K. In the temperature range of 278–318 K, the total rate constants (*k*_total_ and *k*′_total_) of the reactions of MNZ with HO• and SO_4_^•−^ decreased as the temperature increased. The total addition rate constants (*k*_total−RAF_ and *k*′_total−RAF_) had negative temperature dependence. However, the total hydrogen abstraction rate constants (*k*_total−HAT_ and *k*′_total−HAT_) had weakly positive temperature dependence. It is worth noting that the total rate constant was mainly determined by the radical adduct formation channel. As shown in Table 1, the total rate constants of radical adduct formation channels of the reactions of MNZ with HO• and SO_4_^•−^ were 8.10 × 10^9^ and 1.69 × 10^9^ M^−1^ s^−1^, accounting for 95% and 100% for the corresponding total reactions. It was indicated that hydrogen abstraction reactions can be neglected and radical adduct formation reactions are the most important during the aqueous phase. The above kinetic calculations are consistent with the conclusions obtained from thermodynamics.

The half-life (*t*_1/2_) of the OH-initiated transformation of MNZ can be calculated by the total rate constant and the concentration of HO•: *t*_1/2_ = ln2/(*k*_aq_ × [HO•]). Here, *k*_aq_ is the total rate constant for the reaction of MNZ with HO•. [HO•] is the concentration of HO•. Previous studies have proven that [HO•] reached 10^−9^ to 10^−10^ mol/L in wastewater subjected to AOP treatment and 10^−14^ to 10^−18^ mol/L in natural water [56,57,58]. Thus, the half-life of MNZ initiated by HO• in the temperature range of 278–318 K, with the [HO•] range of 10^−9^ to 10^−18^ mol/L, was calculated and given in Appendix A. The half-life (*t*_1/2_) in AOP wastewater (10^−9^−10^−10^) was 0.081−0.81 s at 298 K (see Appendix A). When the concentration of HO• was lower than 5.88 × 10^−17^ mol/L, MNZ remained in the water for more than 16 days, which indicated that the half-life of MNZ was longer in natural water. Obviously, at a certain temperature, *t*_1/2_ increases with the decrease in the concentration of HO•. At the highest [HO•] of 10^−9^ mol/L, *t*_1/2_ increased from 0.052s to 0.119 s when the temperature increased from 278 K to 318 K. Obviously, at a certain [HO•], *t*_1/2_ increases with the increase in temperature. Therefore, both the temperature and the concentration of HO• influence *t*_1/2_. Briefly, the stay time of MNZ in AOPs is very short, but it persists a long time in natural water.

### 3.3. Risk Predictions

Ecotoxicity assessment plays an important role in controlling organic contaminants. Bai et al. assessed the ecotoxicity of ciprofloxacin (CIP) and its products through ecological structure–activity relationships (ECOSAR). It was found that most of stable degradation products were harmless to the three tested aquatic organisms (fish, daphnia, and green algae) [59]. Recently, the ecotoxicity of mevinphos and monocrotophos and their degradation products were assessed by Tang et al. using the ECOSAR program. The ecotoxicity predictions for the three aqueous organisms implied that their toxicity levels were reduced during degradation [60]. To explore the potential risks of MNZ and its degradation products for aquatic organisms, the acute and chronic toxicities were evaluated using three different nutritional levels of aquatic organisms (daphnia, fish, and green algae) based on the ECOSAR program. The specific classification criteria for acute and chronic toxicity are shown in Appendix A. The predicted toxicity values of MNZ and its degradation products are shown in Appendix A. The acute toxicity values of MNZ to fish, daphnia, and green algae were 878.33, 179.75, and 6.92 mg L^−1^, respectively. It is indicated that MNZ is not harmful to either fish or daphnia, but is toxic to green algae in terms of acute toxicity. The chronic toxicity values of MNZ were 0.95, 3.08, and 2.70 mg L^−1^ for fish, daphnia, and green algae, respectively, showing that MNZ was chronically toxic to fish and harmful to daphnia and green algae. It is indicated that MNZ is a toxic pollutant. Thus, it is necessary to pay further attention to it and its degradation products for aquatic organisms.

As shown in Appendix A, the acute toxicity values of the initial degradation product of M-P were 260.97, 79.78, and 3.56 mg L^−1^ for fish, daphnia, and green algae, respectively. And the chronic toxicity values of M-P were 0.50, 1.37, and 1.45 mg L^−1^ for fish, daphnia, and green algae, respectively. The toxicity evolution of MNZ and its major degradation products is shown in Figure 6. The toxicity level of the initial degradation product of M-P was increased for fish, daphnia, and green algae. The toxic predication for DMZ and ONZ degradation also showed that the initial degradation products of D-P and O-P enhanced aquatic toxicity. The toxicity levels of P1, P2, and P3 were decreased and found not to be harmful to three aquatic organisms. Similarly, in the study of DMZ and ONZ with HO•, it was indicated that most of the secondary degradation products were not harmful to aquatic organisms. Thus, the risk of degradation of MNZ should be a concern in the formation process of M-P.

## 4. Conclusions

In the present research work, the degradation process of MNZ was systematically described from three aspects: the degradation mechanism, the kinetics, and a toxicity evaluation from theoretical point of view. Several findings were concluded:(1)The main reaction channel for both HO• and SO_4_^•−^-initiated MNZ was the RAF reaction. The main reaction site was the C1 atom on the imidazole ring of MNZ. The main by-products obtained in the subsequent hydroxylation reaction were P2 and P3.(2)The total rate constants of MNZ with HO• and SO_4_^•−^ were determined to be 8.52 × 10^9^ and 1.69 × 10^9^ M^−1^ s^−1^ at 298 K in an aqueous environment, which is in agreement with the experimental values.(3)The toxicity analysis showed that the end products were not harmful to aquatic organisms. The toxicity level of the initial degradation product of M-P was higher than the parent of MNZ.

We are finally able to answer the questions surrounding MNZ degradation. These results suggest that HO• and SO_4_^•−^ oxidation is effective in removing MNZ from aquatic environments. This study can guide the aquatic transformation and removal of nitroimidazole antibiotics.

## Figures and Tables

**Figure 1 toxics-11-00796-f001:**
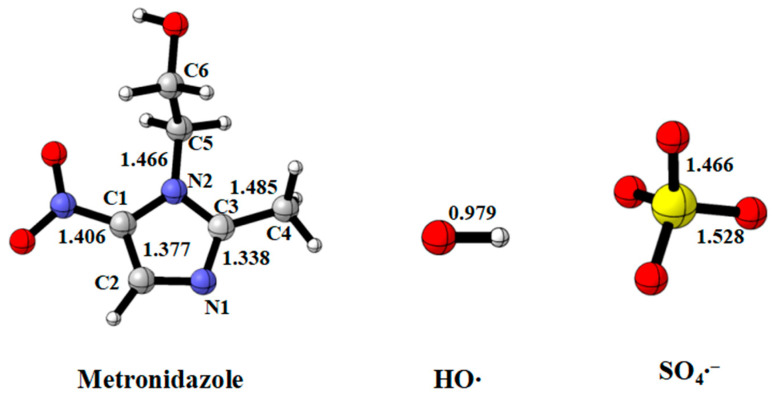
Optimized structures of MNZ, HO•, and SO_4_^•−^ at the M06-2X/6-31+G(d, p) level. Bond lengths are given in angstroms 
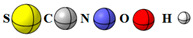
.

**Figure 2 toxics-11-00796-f002:**
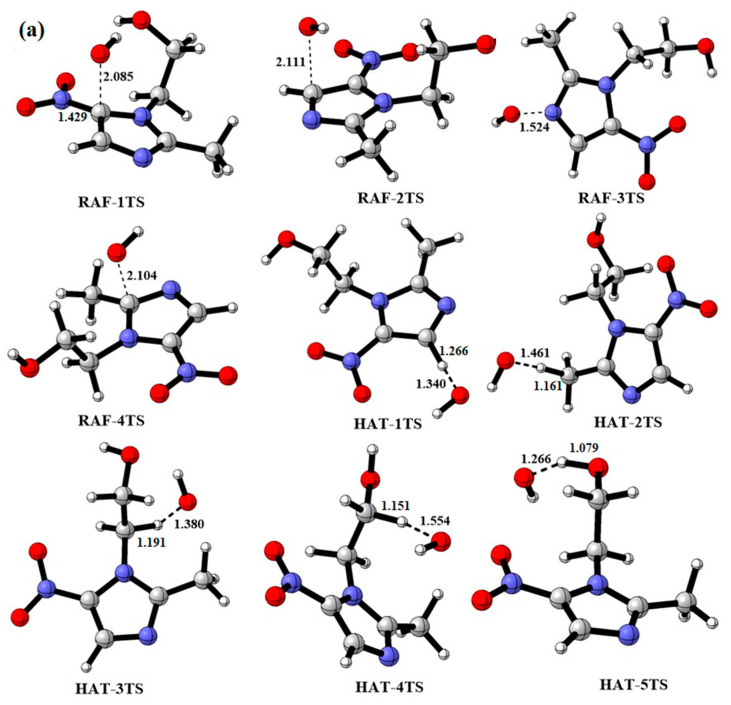
Optimized geometries of the transition states (TSs) involved in the initial reactions of MNZ with HO• and SO_4_^•−^ (**a**) MNZ with HO• reactions; (**b**) MNZ with SO_4_^•−^ reactions. Bond lengths are given in angstroms 
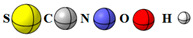
.

**Figure 3 toxics-11-00796-f003:**
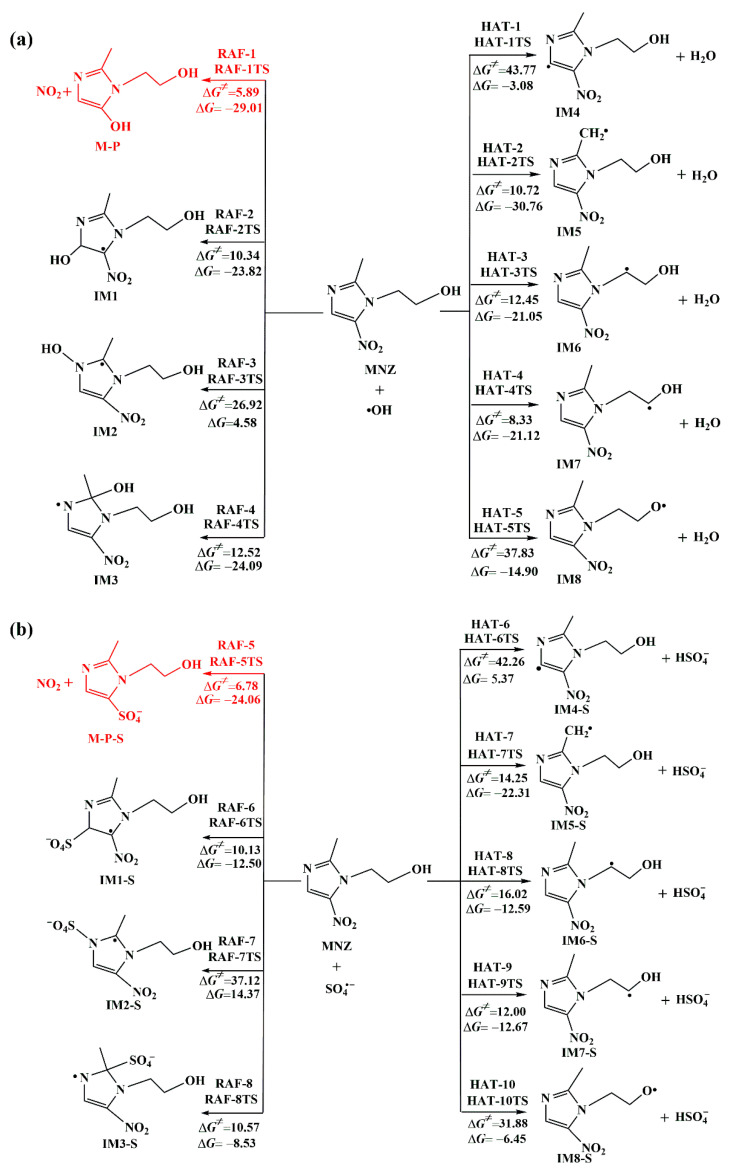
The detailed initial reaction pathways of MNZ with HO• and SO_4_^•−^ including Gibbs free energy of activation (Δ*G*^≠^) and Gibbs free energy of reaction (Δ*G*). All values are in kcal mol^−1^. (**a**) MNZ with HO•; (**b**) MNZ with SO_4_^•−^. The lowest energy barriers pathways are marked in red.

**Figure 4 toxics-11-00796-f004:**
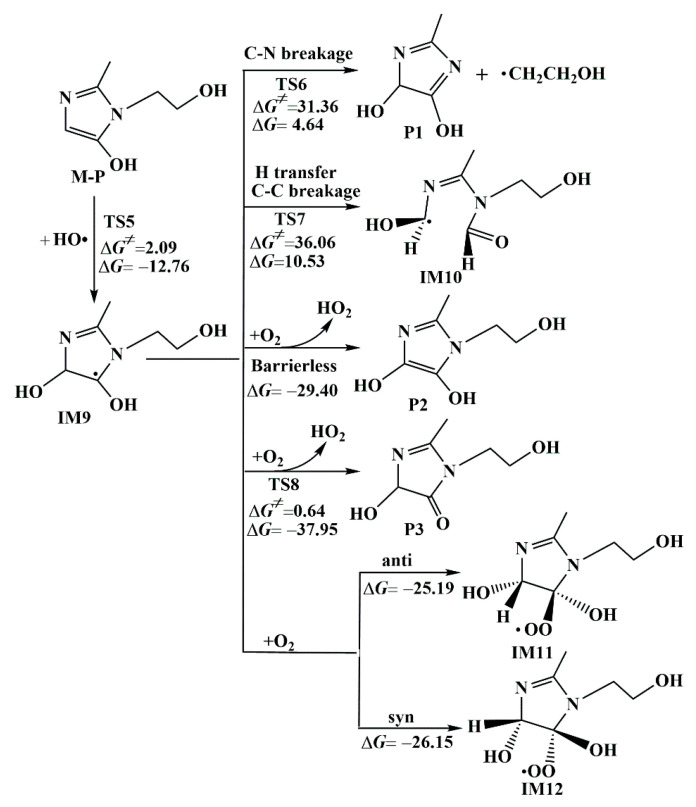
Subsequent reactions of M-P with Gibbs free energy of activation (Δ*G*^≠^, kcal mol^−1^) and Gibbs free energy of reaction (Δ*G*, kcal mol^−1^).

**Figure 5 toxics-11-00796-f005:**
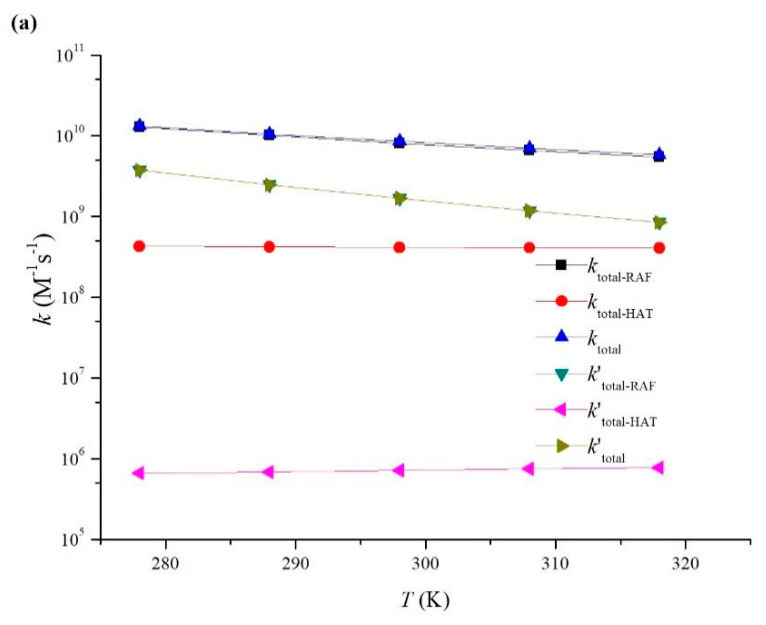
The calculated total rate constants (*k*_total_ and *k*′_total_), the total addition rate constants (*k*_total−RAF_ and *k*′_total−RAF_), and the total hydrogen abstraction rate constants (*k*_total−HAT_ and *k*′_total−HAT_) for the corresponding reactions of MNZ with HO• and SO_4_^•−^ in the temperature range of 278–318 K. (**a**) *k* vs. *T*; (**b**) *lnk* vs. 1/*T.*

**Figure 6 toxics-11-00796-f006:**
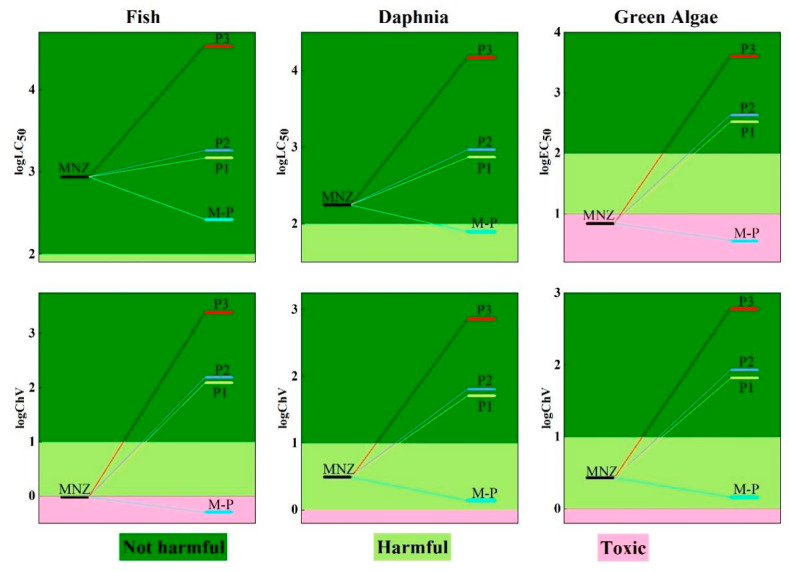
Acute and chronic toxicity values of MNZ and its main degradation products.

**Table 1 toxics-11-00796-t001:** The rate constants and branching ratios for the initial reaction channels of MNZ with HO• and SO_4_^•−^ at 298 K.

Reactions	HO•	Reactions	SO_4_^•−^
*k* _aq_	*Г* (%)	*k* _aq_	*Г* (%)
RAF-1	8.10 × 10^9^	95	RAF-5	1.68 × 10^9^	99.5
RAF-2	4.49 × 10^6^		RAF-6	5.78 × 10^6^	0.3
RAF-3	4.32 × 10^6^		RAF-7	3.35 × 10^−13^	
RAF-4	1.36 × 10^5^		RAF-8	2.72 × 10^6^	0.2
*k* _total-RAF_	8.10 × 10^9^	95	*k′* _total-RAF_	1.69 × 10^9^	100
HAT-1	2.44 × 10^−18^		HAT-6	6.08 × 10^−17^	
HAT-2	1.38 × 10^7^		HAT-7	2.80 × 10^4^	
HAT-3	6.46 × 10^5^		HAT-8	1.20 × 10^3^	
HAT-4	3.98 × 10^8^	5	HAT-9	6.85 × 10^5^	
HAT-5	1.22 × 10^−13^		HAT-10	2.70 × 10^−9^	
*k* _total-HAT_	4.13 × 10^8^	5	*k′* _total-HAT_	7.15 × 10^5^	
*k* _total_	8.52 × 10^9^		*k′* _total_	1.69 × 10^9^	
*k* _experiment_	(3.54 ± 0.42) × 10^9^; (5.5 ± 0.2) × 10^9^		*k′* _experiment_	(2.74 ± 0.13) × 10^9^

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
