# Peer review of "A Theoretical Study on the Degradation Mechanism, Kinetics, and Ecotoxicity of Metronidazole (MNZ) in •OH- and SO4•−-Assisted Advanced Oxidation Processes"

_toxics, 2023, doi:10.3390/toxics11090796_

Round 1

Reviewer 1 Report

This manuscript by Jingyu Sun and coauthors deals with the degradation mechanism, kinetics and ecotoxicity of metronidazole in •OH and SO4•−-assisted advanced oxidation processes. A theoretical study of the electronic structure and kinetic calculations within the Gaussian 16 software program with the more expensive 6-311++G(3df, 2p) basis set, as well as intrinsic reaction coordinate (IRC) calculations, KiSThelP program, Ecological Structure-Activity Relationship Model (ECOSAR 96 V2.0), etc.

The manuscript is well organized and well written. The results support the conclusions, however, there are a few queries listed below. In particular, the authors identified the main channel for both HO• and SO4•−-initiated metronidazole, which is relevant to the main by-products. The toxicity results of the initial degradation product are found to be higher than metronidazole. The new results are found for the acute toxicity values of metronidazole to fish, daphnia and green algae.

The work can be published in Toxics after the following queries that the authors should correct before publication: 

(1) In Eq. 2.1, the other parameters should be specified in the text

(2) Figure 4 is not cited and discussed in the text

(3) Section Conclusions contain a part of conclusions, as compared with Abstract and the text, also prospects should be added

Minor changes are required

Author Response

Reviewer #1:

  1. In Eq. 2.1, the other parameters should be specified in the text.

Answer: Many thanks for the suggestion. All parameters have been specified in the text in Eq. 2.1.

  1. Figure 4 is not cited and discussed in the text.

Answer: Figure 4 has been cited and discussed in the text.

  1. Section Conclusions contain a part of conclusions, as compared with Abstract and the text, also prospects should be added.

Answer: The prospects have been added in the conclusion.

Reviewer 2 Report

the respective file is attached

Moderate editing of English language required

Author Response

Reviewer #2:

  1. From the content of Introduction, it is immediately seen that author(s) are either not familiar with the literature related to the AOP based on gamma-ray, X-ray and electron beam from the electron accelerators or for some unknown reason they totally neglected this very important technique broadly applied in decomposition of various pollutants in waters and waste waters. Authors are strongly recommended to see the recent review by Trojanowicz et al. published as a chapter "Gamma-ray, X-ray and Electron beam Based Processes”in Advanced Oxidation Processes for Wastewater Treatment, S. Ameta, R. Ameta (eds), Elsevier, 2018, 257-331. In connection with MNZ authors should also get acquainted with the paper by M. Sanchez-Polo et al. in Water Research, 2009, 43, 4028-4036.

Answer: Many thanks for the suggestion. The two references have been cited in the introduction.

  1. Authors limited the role of MNZ as a kind of anti-anaerobic and anti-protozoal antibiotic and an additive in poultry and fish feeds. Again they totally neglected another potential role of MNZ as an effective radiosensitizer which can be used in radiotherapy to overcome the radio-resistance of hypoxic tumor cells. Authors are again strongly recommended to see several important original papers addressing this issue: British Journal of Radiology 1974, 47, 474-481; Int, J, Radiat. Biol. 1974, 26, 557-569, Environmental Health Perspectives 1985, 64, 309-320.

Answer: The relevant three references have been cited in the introduction.

  1. The structure of the manuscript looks nearly like a template of authors earlier paper in Journal of Hazardous Materials 2022, 422, 126930). As an example, see for comparison Figures 3 and 4 for MNZ with the respective Figures 3 for dimetridazole (DMZ) and ornidazole (ONZ) and Figures 5 and Figure 6 (for DMZ and ONZ, respectively). Moreover, what is a novelty in the reaction mechanism involving •OH radicals and products of MNZ degradation and earlier studied nitroimidazoles (DMZ and ONZ) with MNZ? This should be discussed in a more detailed manner.

Answer: The similar reaction mechanisms and degradation products were found in the reactions of •OH with MNZ, DMZ and ONZ. We also discussed the relevant results in the text. The reaction feature summarized should be a novelty for reaction mechanism.

  1. Authors should also confront their results obtained for •OHradicals with the results

obtained by Xu et al. J. Phys. Chem. A 2019, 123, 933-942 and critically discuss differences and similarities.

Answer: We have compared our results with the results obtained by Xu et al.

  1. Page 3, lines 122 – 123: Why the numbers for ΔG#for OH and SO4•− addition to the N1 atom differ from those given on Figure 3, 26.37 vs 26.92 and 36.46 vs 37.12? Please explain this discrepancy and correct to the proper ones.

Answer: The values have been corrected.

  1. Page 3, line 125: Why the numbers for ΔG#for OH and SO4•− addition to the C1 atom differ from those given on Figure 3, 5.34 vs 5.89 and 9.47 vs 6.78? Please explain this discrepancy and correct to the proper ones.

Answer: The values have been corrected.

  1. Page 5, lines 135 – 136: Why the numbers for the range of ΔG for H-atom abstraction by OHdiffer from those given on Figure 3, -2.55 to -31.31 vs -2.00 to -30.76? The same applies for the range of the respective ΔG#: 10.17 to 43.22 vs 10.72 to 43,77. Please explain this discrepancy and correct to the proper ones.

Answer: The values have been corrected.

  1. Page 5, line 136: Why the numbers for the range of ΔG for H-atom abstraction by SO4differ from those given on Figure 3: -7.12 to -23.24 vs -6.45 to -22.31. Please explain this discrepancy and correct to the proper ones. By the way, why ΔG for HAT-6 is positive? Is the error or the real value?

Answer: The values have been corrected.

  1. Page 7, lines 165 – 173: This text is totally unclear. Looking at the thermodynamical data formation of P1 and IM10 from IM9 is thermodynamically unfavourable since ΔGs of both reactions are positive, moreover, they are also accompanied with the high Gibbs free energy barriers ΔG#, respectively. If IM10 cannot be formed the consideration of its reaction with oxygen is irrelevant. Moreover, there is also an error (line 172) since this reaction does not lead (according to Figure 4) to product P4 but P2.

Answer: Looking at the thermodynamical data, P1 and IM10 were less likely to be formed. In order to specify reaction mechanism, we listed some possible reaction pathways. The product is P2 which has been revised.

  1. Page 7, lines 165 – 176: why ΔG#for TS8 is negative (-8.63)? Is the error or the real value? Please explain, and correct if necessary.

Answer: It should be barrierless process, but TS8 is the correct transition state.

  1. Page 8, Table 1: Why the calculated rate constant of the reaction RAF-1 with the

most favourable thermodynamic parameters (Figure 3) is lower than the rate constant

of the reaction RAF-2 with higher Gibbs free energy barriers ΔG# 10.34 vs 5.89) and less negative Gibbs free energy ΔG (-23.82 vs. -29.01)? The same applies for the reaction RAF-5 and the reaction RAF-6. On the other hand, authors consider products M-P and M-P-S as the main primary products of reactions of MNZ with •OH and SO4•−, though the rate constants of reactions leading to them are nearly 3-orders of magnitude lower. Can authors comment on that issue?

Answer: Table 1 has been corrected.

  1. Page 9, lines 197 – 200: Can authors comment why the calculated rate constants of MNZ reaction with both radicals (OHand SO4•−) decrease with temperature? These are rather unexpected results since van’t Hoff’s rule predicts the opposite trend. In a consequence, the plot of lnk vs 1/T will give the negative activation energies for these two reactions. In connection with this issue, authors calculated rate constants in the range of 200 K to 400 K. In this temperature range (under normal pressure conditions 1 atm) water undergoes two phase transitions: at 273 K and 373 K. How these transitions were implemented and taken into account in the kinetic calculations?

Answer: The rate constants for the most important addition reaction pathways had negative temperature dependence because the value of their internal energy is negative that lead to negative temperature dependence. The rate constants at 273 K and 373 K have been calculated and listed in Tables S2 and S3.

  1. Page 8, line 190: Authors compare their calculated rate constant for OHwith MNZ with the experimental value measured in ref. 43. They neglected the following reference: Whillans et. al. Radiat. Res. 1975, 62, 407 where this rate constant was measured by means of pulse radiolysis technique and its value (5.5 ± 0.2) × 109 M-1S-1 is even closer to the calculated one.

Answer: The reference has been cited.

  1. Page 9, lines 208 -209: The caption and the legend of Figure 5 is not informative enough. Reader has to check Tables S2 and S3 to assign which plots can be assignedto the reactions of •OH+ MNZ and SO4•− + MNZ, respectively. By the way, there is no reference to Tables 2 and 3 in the main text of the manuscript.

Answer: Figure 5 shows the rate constants of MNZ reaction with HO and SO4•− as a function of temperature from 200 to 400 K. Tables S2 and S3 list rate constants for initial reaction pathways. The trend of the rate constant with temperatures needs to see Figure 5, and the values of rate constants need to see Tables 2 and 3.

Round 2

Reviewer 2 Report

The review is attached in the separate file

As above

Author Response

Reviewers' comments: 

Reviewer #2:

  1. Authors'reply does not satisfy the reviewer. It is not clear what authors means by the reaction feature summarized which according to them is a novelty for reaction mechanism. This novelty should be expressed specifically. On the other hand, authors stated in their response that similar reaction mechanisms and degradation products were found in the reaction of OH with MNZ, DMZ and ONZ. So, what is a novelty in the reaction mechanism found for MNZ? Therefore, I repeat my introductory comment from my previous review: this manuscript is an illustration of the following approach: one new nitroimidazole compound, another new publication. Since the number of known nitroimidazoles is quite significant, it can be expected that the another selected new nitroimidazole will be the subject of the next publication.

Answer: Many thanks for the suggestion. The reaction mechanisms of •OH with MNZ, DMZ and ONZ are greatly similar. However, the study of •OH with MNZ reaction is also very important because the reaction mechanism of MNZ degraded by HO and SO4 can guide other new nitroimidazoles to be removed. Meanwhile,•OH with MNZ reaction mechanism can guide the study of SO4 with MNZ. By calculations, we have also known that the HO• radicals with MNZ, DMZ and ONZ caused the addition to C1 position followed by the products of M-P/D-P/O-P and NO2 with the barriers of 5.89, 4.94 and 6.32 kcal mol−1, respectively. These barrier data can help us to build reaction model for different nitroimidazoles with •OH avoiding a large number of calculations.

  1. The authors did not address my full comment. They did not explain why ΔGfor HAT-6 is positive? Is the error or the real value?

Answer: The ΔG value for HAT-6 is positive because this pathway is endothermic. We calculated and checked this transition state again, and it is right.

  1. I do not agree with the authors to list some possible reaction pathways which according to their own calculations are thermodynamically unfavourable (Figure 4). If they want to list them, anyway, they should cross out the reaction arrows for reaction TS6 and TS7 showing that are potentially possible but thermodynamically not. Therefore, reaction of IM10 with O2is totally irrelevant and should be removed from the Figure 4.

Answer: According to your suggestion, the reaction of IM10 with O2 has been removed from the Figure 4.

  1. Authors removed ΔG#= -8.63 value and replaced it by barrierless. So, the method applied does not properly calculated the activation energy involving TS8. There is no structure of the transition state TS8 available in the manuscript or supplementary material. On what basis authors claim that its structure is correct?

Answer: The ΔG# of TS8 is lower than reactants 8.63 kcal/mol, which indicated that this is a barrierless process. In the IRC calculation of TS8, the products would be formed directly from reactants. Thus, this transition state isn't listed in the Figure 4.

  1. First of all the first sentence of authors'answer is unclear. What the term″…the value of their internal energy is negative…″refers to? From the structure of this sentence it looks that it refers to″addition reaction pathways″. This does not make sense. Authors probably have in their minds the values of Gibbs free enthalpy of reaction substrates, transition states and products? These information should be given in Supplementary Material for checking whether the calculated values are correct.

Answer: The Gibbs free energies (ΔG) and relative energies (ΔE) for intermediates and transition states of MNZ with HO and SO4•− initial reaction have been listed in Table S2.

  1. Secondly, authors did not refer at all to the following comment: In connection with this issue, authors calculated rate constants in the range of 200 K to 400 K. In this temperature range (under normal pressure conditions 1 atm) water undergoes two phase transitions: at 273 K and 373 K. How these transitions were implemented and taken into account in the kinetic calculations? In the range between 200 K and 273 K we are dealing with solid water (ice), on the other hand in the range between 373 K – 400 K with water vapor. By the way, what was the motivation to calculate rate constants for the reaction of OH with MNZ not only in aqueous phase but also in solid and gas phases? It is not true that the rate constants at 273 K and 373 K have been calculated and listed in Tables S2 and S3.

It is very strange that the calculated rate constants for kRAF-2 and in a consequence for ktotal-RAF at temperatures below 273 K are higher (2.09 ×1011 M-1 s-1 to 1.67×1010 M-1 s-1 ) than the rate constants measured experimentally for diffusion-controlled reactions in aqueous phase. In aqueous media, the rate constant 1011 M-1 s-1 was measured only for the neutralization reaction (H++HO ). The rate constant limit for diffusion-controlled reactions measured in aqueous solutions is about 1.5×1010 M-1 s-1. I have serious doubts whether the rate constant values calculated by authors for the solid and gas phases are correct.

Answer: In order to avoid ice and water vapor, we recalculated the rate constants in the temperature range of 278318 K. The relevant results have been shown in Figure 5, Table S4, and Table S5.

  1. Thirdly, though authors corrected the rate constants for RAF-1 and RAF-2 and RAF-5 and RAF-6 calculated for 298 K presented in Table 1, but they left the wrong values of the rate constants for the same reactions calculated in the range 200K – 400K in Tables S2 and S3. This gives a very bad impression and undermines confidence in the calculated rateconstants for the individual reactions. One can get the impression that the authors approach their results completely uncritically.

Answer: Tables S2 and S3 at present Tables S3 and S4 have been revised.

  1. Answer to Comment 14: Figure 5 shows the rate constants of MNZ reaction with HO• and SO4as a function of temperature from 200 to 400 K. Tables S2 and S3 list rate constants for initial reaction pathways. The trend of the rate constant with temperatures needs to see Figure 5, and the values of rate constants need to see Tables 2 and 3. Authors did not refer at all to my all comments. Their answer is completely off topic.

Answer: The caption and the legend of Figure 5 have been revised.

Round 3

Reviewer 2 Report

the review is attached in a separate file

see my recommendatiopn above

Author Response

  1. What authors mean by intermediates, substrates of the respective reactions? The term “intermediates” in this particular case is not clear. This should be corrected in the caption of Table S2. Why four numbers in Table S2 are marked in red?

Answer: The caption of Table S2 has been revised. The intermediate means middle product that is abbreviated as IM. The relative energies (ΔE) of four numbers in Table S2 are negative. Thus, they are marked in red in order to explain negative dependence of rate constant with temperature. The four numbers in Table S2 have been marked in black.

  1. The authors did not respond to my comments at all. They simply cut out all the results for the solid and gas phases. Did they have no arguments to defend or correct their calculations for these two phases? This proves that the authors approached their results in a completely uncritical way.

Answer: In section 3.2, the comments have been added. The aqueous phase temperatures of 278−318 K were considered in order to avoid ice and water vapor. The rate constants in the solid phase are difficult to be calculated because the reactions involved ice are extremely complex and different from reactions in gas and aqueous phase. The transition state theory is no longer applicable to calculate the solid phase rate constant. However, the reaction rate constants involved water vapor could be calculated by TST. Previous studies showed that the kinetic calculations for gas-phase reactions could provide better results [40,49,54]. And gas-phase rate constants could be changed to that of aqueous-phase. However, in this paper, gas-phase rate constants weren't be calculated because the reactions of MNZ with HO• and SO4•− mainly occurred at aqueous phase.   

  1. OK, accepted. However, the reviewer is wondering why authors did not show the calculated rate constant for various temperatures in the form of a typical Arrhenius plot, e. ln k vs 1/T? Reviewer strongly suggests to include such plot as an inset in Figure 5

Answer: A typical Arrhenius plot, i.e. ln k vs 1/T has been put in the Figure 5 as Figure 5(b).